# Auscultation-Based Pulmonary Disease Detection through Parallel Transformation and Deep Learning

**DOI:** 10.3390/bioengineering11060586

**Published:** 2024-06-08

**Authors:** Rehan Khan, Shafi Ullah Khan, Umer Saeed, In-Soo Koo

**Affiliations:** 1Department of Electrical Electronic and Computer Engineering, University of Ulsan, Ulsan 44610, Republic of Korea; rehankhan.mte@gmail.com (R.K.); shafiukhan98@gmail.com (S.U.K.); 2Research Centre for Intelligent Healthcare, Coventry University, Coventry CV1 5FB, UK; saeedu3@uni.coventry.ac.uk

**Keywords:** respiratory sounds, LSTM, mel spectrogram, convolutional autoencoder, artificial intelligence, continuous wavelet transform, hybrid features, healthcare

## Abstract

Respiratory diseases are among the leading causes of death, with many individuals in a population frequently affected by various types of pulmonary disorders. Early diagnosis and patient monitoring (traditionally involving lung auscultation) are essential for the effective management of respiratory diseases. However, the interpretation of lung sounds is a subjective and labor-intensive process that demands considerable medical expertise, and there is a good chance of misclassification. To address this problem, we propose a hybrid deep learning technique that incorporates signal processing techniques. Parallel transformation is applied to adventitious respiratory sounds, transforming lung sound signals into two distinct time-frequency scalograms: the continuous wavelet transform and the mel spectrogram. Furthermore, parallel convolutional autoencoders are employed to extract features from scalograms, and the resulting latent space features are fused into a hybrid feature pool. Finally, leveraging a long short-term memory model, a feature from the latent space is used as input for classifying various types of respiratory diseases. Our work is evaluated using the ICBHI-2017 lung sound dataset. The experimental findings indicate that our proposed method achieves promising predictive performance, with average values for accuracy, sensitivity, specificity, and F1-score of 94.16%, 89.56%, 99.10%, and 89.56%, respectively, for eight-class respiratory diseases; 79.61%, 78.55%, 92.49%, and 78.67%, respectively, for four-class diseases; and 85.61%, 83.44%, 83.44%, and 84.21%, respectively, for binary-class (normal vs. abnormal) lung sounds.

## 1. Introduction

Globally, lung diseases are acknowledged as highly fatal and dangerous, affecting millions of people every year. According to the Forum of International Respiratory Societies (FIRS), respiratory disorders cause almost four million fatalities annually and are among the leading causes of morbidity worldwide [1]. Furthermore, the World Health Organization (WHO) reported that after cardiovascular diseases, respiratory diseases are the second largest contributor to the global disease burden; approximately 10 million people lose their lives to respiratory diseases every year [2]. The diagnostic procedures for respiratory diseases primarily involve auscultation, wherein medical specialists listen with a stethoscope to the sounds as air moves in and out of the lungs [3]. Lung auscultation is among the traditional diagnostic techniques employed [4] by medical specialists to assess the status of respiratory diseases. Crackles and wheezes are the two most frequently heard abnormal lung sounds [5]. These sounds are identified based on their frequency, pitch, energy, intensity, and duration. Wheezes are continuous, high-pitched noises typically occurring in the 400–500 Hz range with a duration longer than 100 ms. Wheezes are generally heard in individuals with asthma and chronic obstructive pulmonary disease [6]. Crackles are discontinuous sounds with a pitch ranging between 100 and 2000 Hz. Crackles are generally heard in patients suffering from heart failure, pneumonia, and bronchitis [7]. Auscultation is cost-effective, easy to apply, and provides essential details about lung conditions and symptoms for a quick diagnosis [8]. However, traditional auscultation with a stethoscope is not infallible, because it depends on the clinician’s expertise and auditory sensitivity. Sometimes, during an examination, this leads to misclassification, even when carried out by an expert physician [9]. Research by Salvatore and Nieman [10] revealed that more than half of the pulmonary sounds were incorrectly identified by medical trainee students in the hospital. Since lung sounds are non-stationary, it is challenging to distinguish them through traditional auscultation techniques. Therefore, there is a need to develop a respiratory disease detection system to ensure more efficient clinical diagnoses.

The 2017 public respiratory sound dataset released by the International Conference on Biomedical Health Informatics (ICBHI-2017) [11] has attracted significant interest among research teams developing automated systems for distinguishing lung sounds. Deep learning (DL) and conventional machine learning (ML) have been utilized in studies over the last decade to address the classification task [12,13,14]. Several attempts have been made to develop algorithms and methods for feature extraction aimed at automatically identifying abnormal lung sounds. Among them, some common feature extraction techniques include spectrograms [15], mel spectrograms [16], wavelet coefficients [17], and the mel-frequency cepstral coefficient (MFCC) [18], as well as a wide range of DL and ML approaches.

Pham et al. [4] extracted various features, including the short-time Fourier transform (STFT) and mel spectrogram. Gairola et al. [19] employed a convolutional neural network (CNN), leveraging mel spectrograms to identify adventitious lung sounds. Bardou et al. [20] utilized the MFCC and traditional ML features (such as local binary patterns) for feature extraction, replacing the CNN model with fully connected layers to train these features, and integrating the output of four CNN models with softmax activation. The authors of [21] optimized an AlexNet pre-trained CNN model, utilizing scalograms to extract a visual representation of the pixel values to accurately detect and classify lung sounds. Tariq et al. [22] developed a model that concatenates three distinct features (a chromagram, the MFCC, and a spectrogram) to classify lung audio samples using ideal CNN models. Similarly, the study in [23] presented various feature extraction techniques to classify different respiratory diseases such as COPD and asthma.

In addition to lung sound analysis, other research has utilized methods such as the wavelet transform and the spectrogram [24], or empirical mode decomposition (EMD) and bandpass filtering for scale selection, as well as processing continuous wavelet transform (CWT)-based scalogram representations with a lightweight CNN for classification of various respiratory diseases. Recent advancements in noninvasive monitoring have led to significant progress in deriving respiratory signals from ECG data. thereby enhancing traditional respiratory sound analysis. Yi and Park [25] demonstrated the derivation of respiratory signals using wavelet transforms directly from the ECG, establishing a foundation for reliable respiratory monitoring without the subject’s awareness. O’Brien and Heneghan [26] presented a comparative examination of methods for extracting respiratory signal extraction approaches from the ECG, highlighting the accuracy and robustness of these techniques across various body postures during sleep. Furthermore, Campolo et al. [27] introduced a novel technique employing EMD to derive respiratory signals from the ECG, showcasing its superior performance in accurately reconstructing respiratory waveforms. This approach offers a dual-modality method that enhances diagnostic capabilities by simultaneously analyzing cardiac and respiratory data. In [28], the authors classified electroencephalogram (EEG) signals using CWT and a long short-term memory (LSTM) model, similar to the study in [29], in which a dual scalogram comprising the Stockwell transform and a CWT scalogram was employed for fault diagnosis in centrifugal pumps. Furthermore, recent studies have explored different ML and DL techniques for binary-class (normal vs. abnormal) classification and multi-class classification of respiratory diseases [30].

In order to achieve improved performance for multi-class and binary classification tasks, Nguyen and Pernkopf [31] developed approaches that include sample padding, feature splitting, an ensemble of CNNs, and a focal loss objective. Acharya and Basu [15] introduced a deep hybrid-based CNN and a recurrent neural network (RNN) framework for detecting respiratory sounds utilizing mel spectrograms. Concurrently, Demir et al. [32] identified four different lung sounds by combining deep CNN features with a linear discriminant analysis and random subspace ensemble classifier. Additionally, to resolve imbalances in the training data, Petmezas et al. [33] employed a model combining a CNN with LSTM networks that include the focal loss function. The respiratory sound cycle was transformed into a time-frequency representation and processed using the CNN.

In this study, we propose a hybrid DL technique with signal processing techniques for detecting various lung disorders. We introduce parallel transformation for rich features using a parallel convolutional autoencoder (CAE). Initially, the auscultation recordings undergo preprocessing through segmentation of respiratory cycles, followed by a padding technique to modify the length of each respiratory cycle to a fixed size. The respiratory cycle audio signal is transformed into a time-frequency representation using CWT and a mel spectrogram. Two parallel CAEs extract rich features from scalograms, concatenate features in a hybrid pool, and subsequently feed them into an LSTM model that indicates different respiratory diseases.

Our principal contributions are as follows:(1)We present a novel method that combines deep learning and signal processing for enhanced lung auscultation analysis and classification. This approach addresses the limitations of traditional techniques utilized for lung auscultation.(2)This approach utilizes parallel transformation using both CWT and a mel scalogram. A parallel CAE is utilized to extract rich features from the scalograms transformed by CWT and mel at latent spaces.(3)A hybrid feature pool is created by fusing the features collected from both the CWT and mel scalograms using CAE latent spaces. These latent spaces provide an extensive and enriched representation of lung sound features, enhancing the analysis and classification approach.(4)An LSTM network is employed to classify various lung sounds, leveraging its proficiency in handling time-series data. Lung sounds are sequential, and LSTM is particularly suited to recognizing complex patterns and handling sequential information in time-dependent data.

The rest of this paper is organized as follows. Section 2 provides background information on the dataset and comprehensive details of the proposed model. Section 3 describes the experimentation and the model’s performance. Finally, Section 4 summarizes the proposed study along with future expansion and enhancement planned for this work.

## 2. Materials and Methods

The framework of the proposed study for lung sound classification utilizes a hybrid model that combines an autoencoder with a recurrent neural network, specifically the LSTM variant, as illustrated in Figure 1. Initially, all lung sounds are preprocessed to segment the respiratory cycles, ranging from 0.2 s to 16 s, with an average duration of 2.7 s. The respiratory cycles in the dataset are not equal in length, so to address this issue, a padding technique is utilized. Each cycle is preprocessed until the total length equals six seconds. Following this, the cycles are transformed into a dual time-frequency domain using CWT and the mel spectrogram to provide distinct representations of each cycle. Subsequently, these time-frequency spectrogram images are fed into parallel CAEs for feature extraction, and the resulting latent spaces of the parallel CAEs are fused into a hybrid feature pool. Finally, the resulting features from the latent spaces are used as input for the LSTM model to classify various types of respiratory diseases.

### 2.1. Dataset

In this study, the publicly available ICBHI-2017 respiratory sound dataset from the International Conference on Biomedical Health Informatics [11] was utilized. The dataset was collected at two different hospitals in Greece and Portugal by teams of experts. The data were acquired using digital stethoscopes (the AKG-C417 L Microphone, the 3M Littmann Classic II SE, the 3M Littmann 3200, and the Welch Allyn Meditron Master Elite), which have different sampling frequency ranges of 4, 10, and 44.1 kHz. The dataset comprises annotated respiratory cycle recordings totaling 5.5 h. In total, 920 audio samples were collected at various anatomical locations from 126 individuals [34]. The recordings were obtained from healthy individuals and others with a range of pathological conditions, including seven lung diseases: pneumonia, a lower respiratory tract infection (LRTI), asthma, bronchiectasis, an upper respiratory tract infection (URTI), bronchiolitis, and COPD. Furthermore, all the respiratory cycles were annotated based on the presence of crackles and/or wheezes [35]. Wheezes are a type of abnormal, continuous, high-pitched breathing sound primarily associated with chronic disease. In contrast, crackles are discontinuous lung sounds of shorter duration, heard during both the inspiratory and expiratory phases. The duration is notably shorter in the total respiratory cycle and is mainly associated with non-chronic diseases [36].

### 2.2. Preprocessing and Data Augmentation

Each respiratory cycle was annotated by an expert as belonging to one of four classes: normal (N), crackle (C), wheeze (W), and both crackle and wheeze (B). The start times, end times, number of crackles, and number of wheezes are shown in Table 1. The source database contains 6898 respiratory cycles, including 1864 with crackles, 886 with wheezes, 3642 with no labels (i.e., from healthy individuals), and 506 with both crackles and wheezes, as shown in Table 2. The duration of the respiratory cycles is not fixed. Although training DL models is possible by utilizing adaptive average pooling, this approach performs poorly in comparison with a fixed-size signal [19]. The length of the audio signals in the dataset varies, so zero padding was employed to achieve a fixed duration of six seconds. Padded samples of respiratory cycles are shown in Figure 2. Data augmentation techniques were utilized to artificially expand the unbalanced dataset by modifying the audio samples, resulting in several modified versions of the dataset, as shown in Table 3.

The time-domain audio data augmentation approaches employed to enlarge our audio samples were as follows:(1)Time Stretching: This technique involves either increasing or decreasing the sample speed by specific factors [37]. In this work, we augmented the minority class by stretching the respiratory audio signals along with their temporal variations at a stretching rate of 1.2. The length of an audio signal was adjusted based on this rate, calculated by multiplying the original length of the audio by the stretching rate. The time stretching method is useful for modifying the audio’s temporal properties without altering its pitch, which makes it effective for enhancing datasets that have a low representation of particular class samples. A more balanced dataset with improved temporal diversity in the audio signals is the anticipated outcome.(2)Pitch Shifting: This technique involves modifying the lung sound signal by increasing or decreasing the pitch while keeping the audio signal duration constant. In [38], the significance of the pitch-shifting process was investigated for CNN-based sound classification. To enlarge minority classes in the dataset, we employed pitch shifting by randomly shifting the audio signals along the time axis by a maximum percentage value of 0.2. Pitch shifting changes the audio’s frequency content, adding additional variances that enhance the model’s ability to generalize. An expanded range of pitch-modified samples is anticipated, which will strengthen and balance the training dataset.(3)Adding Noise: To further augment the dataset, noise was added to the recordings to increase the sample sizes of minority classes. Noise was introduced from within the function, and a noise vector was generated using a Gaussian distribution of zero mean and unit variance with a length matching the input audio signal. By scaling this noise vector by a factor of 0.005, the amplitude of the noise could be controlled to achieve the desired augmentation. The scaled noise vector was then added element-wise to the original signals, resulting in augmented samples. Using this noise-adding technique effectively enhances the model’s resistance to noise and other fluctuations found in real-world recordings.

### 2.3. Transformation of Lung Sounds

During preprocessing, time-frequency analysis is performed to transform the audio sample into a parallel scalogram. Instead of directly feeding the audio signals into the classification model, we first transform them into a spectrogram from the time-series domain to the time-frequency domain. Transformation is a crucial technique for transforming the audio lung samples into a time-frequency domain, specifically into parallel spectrograms. STFT is applied to the time-domain signal, S(τ), to compute the spectrogram using Equations (1) and (2), where *t* denotes the time localization and W(τ−t) is the window function that cuts and filters the signal [22]. The angular frequency is denoted by ω, and *j* is the imaginary unit, defined as the square root of −1. This process facilitates detailed analysis of lung sound signals, providing a comprehensive feature set for the subsequent classification task.
(1)$$Spectrogram(t,w)={\left|STFT\left(t,w\right)\right|}^{2}$$
(2)$$STFT(t,w)=\int_{-\infty }^{\infty }S\left(\tau \right).W\left(\tau -t\right).e^{-j\omega \tau }d\tau$$

#### 2.3.1. Mel Scalogram

The mel spectrogram, the human auditory system, and scientific research on speech processing are the sources of inspiration for the mel scale. The human ear is more sensitive to variations in lower frequencies than in higher ones and perceives loudness on a logarithmic scale as opposed to a linear one. Transforming the lung sound sample using STFT converts the signal from the time domain to the frequency domain at a sampled frequency of 4000 Hz. A two-dimensional (2D) image is generated, where columns represent time (windows) and rows represent frequencies in the mel scale. Each value in the image corresponds to the signal’s log amplitude for a specific frequency and set of time windows. The time domain is transformed into the frequency domain via STFT. Then, the frequency is mapped to the mel scale and the color dimension to the amplitude [39]. Equation (3) is used for calculating the mel scale, where *f* represents the frequency:(3)M=2595log(1+f/700)

We obtain the log mel spectrogram after computing the logarithm values to condense the dynamic range. The mel spectrogram provides an intricate representation of the power spectrum, showing the energy distribution across frequencies over time. The log value of the energy is expanded in the time domain to generate the mel spectrogram. Figure 3a, b, respectively, illustrate the respiratory cycle sound signal and the mel spectrogram of a respiratory cycle.

#### 2.3.2. CWT Scalogram

CWT is an effective technique for signal processing and is used for analyzing non-stationary signals, including audio signals. Within the context of respiratory sound analysis, CWT provides a robust method for extracting relevant features that capture variations in frequency content over time. Respiratory sounds are recorded using various stethoscopes, producing non-stationary signals. The wavelet transform preserves temporal resolution and computationally analyzes non-stationary signals by decomposing them into different frequency components. The wavelet transform utilizes fundamental operations known as wavelets, enabling simultaneous analysis in both frequency and time domains. The mathematical expression for the wavelet transform is shown in Equation (4):(4)$$WT\left(s,t\right)=\frac{1}{\sqrt{\left|s\right|}}\int_{-\infty }^{\infty }f\left(\tau \right)\psi^{*}\left(\frac{\tau -t}{s}\right)d\tau$$
where f(τ) represents the time-frequency domain of the input signal, $\psi^{*}\left(\cdot \right)$ is the conjugate of the wavelet function scaled by factor *s*, and the translation factor correlating with the time adjustments is denoted by *t*, where the scale factor s > 0. The multi-resolution capabilities of CWT are particularly advantageous for deciphering time-frequency signals since various physiological events may manifest at various scales. The mathematical representation of the CWT details the relationship between the wavelet ψ(t) and function ψ(t), as shown in Equation (5).
(5)$$CWT_{f}\left(s,t\right)=\int_{-\infty }^{\infty }f\left(\tau \right)\psi_{s,t}^{*}\left(\tau \right)d\tau$$

A complex Morlet wavelet is employed as a mother wavelet, ψ(t), in Equation (6):(6)$$\psi_{s,t}\left(\tau \right)=\frac{1}{\sqrt{\left|s\right|}}\psi \left(\frac{\tau -t}{s}\right)d\tau$$
where *s* is the scaling factor and *t* is the translation factor that adjusts the function in time, determining whether it is stretched or compressed, depending on whether *s* > 1 or 0 < *s* < 1. The normalizing term, $\frac{1}{\sqrt{\left|s\right|}}$, ensures that the wavelet energy remains constant across all scales. Equations (4) and (5) are utilized to translate and scale the original mother wavelet, τ, for analysis of a signal at various frequencies and time positions. CWT converts the lung audio signal into images using the Morlet wavelet.

In fact, a complex sinusoid with Gaussian windows forms the complex Morlet wavelet, and the wavelet transform’s best time localization is achieved via its second-order exponential decay. Moreover, the complex Morlet wavelet function is particularly suited to capturing coherence between harmonic frequencies, providing information on both amplitude and phase. CWT with the Morlet wavelet as the mother wavelet allows for the extraction of detailed images of the lung sound wave spectrum, demonstrating temporal resolutions. Figure 4a and b, respectively, show the respiratory cycle audio signal and a CWT image of the respiratory cycle.

### 2.4. Convolutional Autoencoders

CAEs have garnered significant attention in recent years owing to their ability to learn hierarchical representations of data, particularly in image processing tasks. Initially introduced by Theis et al. [40] and Ballé et al. [41], CAEs are specialized neural networks designed to encode and decode spatially hierarchical inputs such as images. CAEs use convolutional layers to leverage spatial locations in data, making them particularly adept at processing images. The primary goal of a CAE is to approximate an identity function while abiding by particular limitations, such as hidden layers having a certain number of neurons. In a CAE, the encoder functions as a funnel, mapping the input, $x\in R^{n}$, to a latent space. The input consists of *n* feature maps, $x\in R^{n\times l\times l}$, originating from the first layer, where each feature map covers l×l pixels, and the output layer contains *m* feature maps involving convolutional kernels. The dimensions of the convolutional kernel are d×d with d≤1.

The process begins by encoding the input image, which is segmented into d×d pixel patches, labeled as $x_{i}$, where *i* = 1, 2, 3,..., *p*. For each patch, the input image is extracted, and convolution operations are carried out using weight $w_{j}$ of the *j*th convolution kernel, resulting in neuron values $o_{ij}$ for *j* = 1, 2, 3,…, *m* in the output layer:(7)$$o_{ij}=f\left(x_{i}\right)=\sigma \left(w_{j}\cdot x_{i}+b\right)$$

The nonlinear activation function is represented by the symbol σ. In this study, the rectified linear unit (ReLU) activation function is used:(8)$$ReLU\left(x\right)=\left\{\begin{array}{cc} x & \mathrm{if}\;x\geq 0\\ 0 & \mathrm{if}\;x<0 \end{array}\right.$$

After convolutional decoder output $o_{ij}$ is processed through encoding, $x_{i}$ is reconstructed using $o_{ij}$ to obtain ${\hat{x}}_{i}$: (9)$$x_{i}=f^{\prime }\left(o_{ij}\right)=\varphi \left(w_{i}\cdot o_{ij}+\hat{b}\right)$$

The CAE layer is optimized through the iterative refinement of weights and errors using stochastic gradient descent. These optimized parameters are used to create the feature maps. For every instance, ${\hat{x}}_{i}$ is formed after convolutional encoding and decoding. The reconstruction process involves patches *p*, each with a size of d×d, and the mean square error between the reconstructed patch, ${\hat{x}}_{i}$, and the original input picture patch, $x_{i}$, where *i* = 1, 2, 3,…, *p*. Equation (10) presents the cost function in its unique form, while Equation (11) elaborates on the reconstruction error [42].
(10)$$JC\left(\theta \right)=\frac{1}{p}\sum_{i=1}^{p}L\left[x_{i},{\hat{x}}_{i}\right]$$
(11)$$LC\left[x_{i},{\hat{x}}_{i}\right]=\;∥x_{i}-{\hat{x}}_{i}{∥}^{2}\;={∥x_{i}-\varphi \left(\sigma \left(x_{i}\right)\right)∥}^{2}$$

In the proposed model, optimization techniques like backpropagation are used to minimize loss when a multiple-layer encoder and decoder are employed to train a CAE. The classification model takes the scalogram as input and passes it through a series of convolutional layers. These layers isolate key features by gradually reducing the dimensionality of the image. Following encoding, the model undergoes a decoding stage to reconstruct the image into its original state. Table 4 outlines the CAE layers, along with their respective input and output dimensions, as applied in this study. The efficiency of the CAE is shown by high peak signal-to-noise ratio values for CWT and mel at 56.66 dB and 71.01 dB, respectively, indicating accurate image reconstruction. The layers of the working architecture are depicted in Figure 5.

### 2.5. Long Short-Term Memory

LSTM was first proposed in 1997 by Hochreiter and Schmid Huber, and the improved RNN model has gained substantial interest for time-series data owing to its specialized cellular architecture [43].

Typically, an LSTM architecture consists of an input gate, an output gate, a forget gate, and a memory cell. The forget gate initially determines which informational segments the cell states should discard, and it is expressed mathematically as follows:(12)$$f_{t}=\sigma \left(W_{f}\times \left[h_{t-1},x_{t}\right]+b_{f}\right)$$
where $x_{t}$ is the current input; $h_{t-1}$ is the previous hidden layer output; *W* and *b* represent the weight matrix and bias, respectively; and σ is the sigmoid activation. The input gate subsequently controls the retention of data in the cell state by dividing them into two parts, determining which data need to be updated, and configuring the updated state. The following are the mathematical expressions:(13)$$i_{t}=\sigma \left(W_{i}\times \left[h_{t-1},x_{t}\right]+b_{i}\right)$$
(14)$${\tilde{C}}_{t}=\tanh\left(W_{c}\times \left[h_{t-1},x_{t}\right]+b_{c}\right)$$

The output gate plays a crucial role in deciding the final output. The segments of the cell state for the output are determined by the sigmoid function, followed by pointwise multiplication with the output of the tanh function:(15)$$o_{t}=\sigma \left(W_{o}\times \left[h_{t-1},x_{t}\right]+b_{o}\right)$$
(16)$$h_{t}=o_{t}\times \tanh\left(C_{t}\right)$$

In the field of biomedicine, LSTMs have shown the capability to recognize time-based patterns, which is particularly useful for the diagnosis of respiratory diseases characterized by detailed time-based patterns. In this study, the LSTM model employs 64 units to process the time-based patterns of the respiratory sound data. Subsequently, the data pass through a dense layer that utilizes a softmax activation function to categorize the LSTM output into specific categories. The model is optimized for best categorization results using the categorical cross-entropy loss function and the “Adam” optimizer. The LSTM architecture employed in this research is detailed in Table 5.

## 3. Results and Discussion

In this study, a publicly available dataset of respiratory sounds was chosen to evaluate the performance of the proposed framework [11]. The proposed DL framework was developed and implemented in Python 3.9.18, leveraging TensorFlow 2.15.0 as the foundation for the Keras library. All experiments were conducted using a desktop computer with an AMD Ryzen 9 5900X 12-Core 3.70 GHz CPU, 64 GB of RAM, and an NVIDIA GeForce RTX 3080 GPU with 64 GB of memory. The respiratory sound dataset encompasses four sub-tasks, which include a binary-class problem distinguishing between normal (N) and abnormal (Ab) samples. Three-class and four-class tasks categorize respiratory cycles into one of four classes (W, C, N, and B). Eight-class categorization is also performed, where classifications include healthy samples and seven distinct lung diseases: pneumonia, LRTI, asthma, bronchiectasis, URTI, bronchiolitis, and COPD. The dataset was split, allocating 80% for training and 20% for testing. After evaluating the proposed model for the binary-class problem, the experimentation was extended to the three-class, four-class, and eight-class problems. We used several metrics to evaluate the performance of respiratory sound classification: accuracy, F1-score, precision, and sensitivity. These metrics collectively provide a nuanced view of the model’s ability to correctly identify and differentiate between the various respiratory diseases. In the classification framework, true positive (TP) is when an instance was accurately identified as positive, and true negative (TN) means an instance was accurately identified as negative. A false positive (FP) is an instance incorrectly identified as positive, and a false negative (FN) is a positive instance incorrectly labeled as negative. The following equations are used to calculate these metrics: (17)$$\mathrm{Accuracy}=\frac{TP+TN}{TP+TN+FP+FN}$$
(18)$$F1\text{-}\mathrm{score}=\frac{2\times (\mathrm{precision}\times \mathrm{recall})}{\mathrm{precision}+\mathrm{recall}}$$
(19)$$\mathrm{Precision}=\frac{TP}{TP+FP}$$
(20)$$\mathrm{Sensitivity}=\frac{TP}{TP+FN}$$

In this study, the proposed model, based on a hybrid approach involving digital signal processing and DL, was evaluated using various classification tasks to assess its effectiveness in distinguishing various respiratory diseases. This evaluation was conducted across multiple classification tasks ranging from simple binary classification problems to three-class, four-class, and eight-class problems. The following are the specific scenarios for each classification problem:Binary-class problems: N-Ab, C-W, B-C, B-W, C-N, C-W, and W-N.Three-class problems: B-C-W, and C-N-W.Four-class problems: C, W, N, and B.Eight-class problems: Healthy (H), pneumonia (P), LRTI (L), asthma (A), bronchiectasis (B1), URTI (U), bronchiolitis (B2), and COPD (C).

### 3.1. Binary Classification

In the binary classification problems, the proposed model demonstrated remarkable accuracy in identifying crucial respiratory sounds. Our model exhibited remarkable performance in differentiating between C, W, N, and B. Several experiments were conducted on both the official and the augmented datasets to validate the effectiveness of our proposed model. For the N-Ab problem, our model achieved an average accuracy of 85.61%, an F1-score of 84.21%, a precision of 85.36%, and a sensitivity of 83.44%. Similarly, for the B-C problem, the results were 94.41%, 93.65%, 93.57%, and 93.74% for accuracy, F1-score, precision, and sensitivity, respectively. For the C-W problem, the results were 93.57%, 93.51%, 93.50%, and 93.53%, respectively. The results for the remaining binary-class problems are shown in Table 6. Figure 6 depicts the confusion matrices, showing the predicted versus the true labels for different binary-class problems.

### 3.2. Three-Class Classification

After achieving promising results for the binary-class problems, we extended our evaluation to three-class classification problems. We further examined and compared the internal relationships and variations for the B-C-W and C-N-W problems. On the official and augmented datasets, our proposed model achieved an average accuracy of 89.45%, an F1-score of 88.41%, a precision of 88.68%, and a sensitivity of 88.16% for the B-C-W problem. For the C-N-W problem, the results were 82.04%, 82.15%, 81.94%, and 82.41%, respectively, as shown in Table 7. Figure 7 presents the confusion matrices for the B-C-W and C-N-W three-class problems.

### 3.3. Four-Class Classification

To evaluate the model’s ability to identify four-class respiratory sound problems, both datasets were used to compare the C, W, N, and B categories. The model demonstrated promising performance across all scenarios, as illustrated by the confusion matrix in Figure 8a. The proposed model achieved an average accuracy of 79.61%, an F1-score of 78.67%, a precision of 78.86%, and a sensitivity of 89.56% on the augmented dataset, as shown in Table 8.

### 3.4. Eight-Class Classification

Finally, the evaluation of the proposed framework for eight-class problems included healthy samples and seven distinct lung diseases (P, L, A, B1, U, B2, and C), as shown in Table 9. The confusion matrix in Figure 8b illustrates that the model yielded an overall accuracy of 94.16%, a sensitivity of 89.56%, an F1-score of 89.56%, and a precision of 89.87%. In summary, these findings demonstrate the proposed model’s robust and reliable performance across various respiratory sound classification scenarios. including binary-class, three-class, four-class, and eight-class problems, even on unbalanced datasets.

### 3.5. Discussion

We proposed a novel approach to evaluating various adventitious lung sounds by employing a hybrid model that combines parallel CAEs and an LSTM network. The model’s performance was evaluated across multiple classification problems: binary-class, three-class, and four-class problems, as well as eight-class problems involving healthy samples and seven distinct diseases. In this study, lung sound signals were not directly fed into the classification model—all lung sound signals were transformed into the frequency domain as spectrograms. For feature extraction, dual CWT and mel transformations were fed into parallel CAEs, and the features extracted from CAE latent spaces were concatenated to create a hybrid feature pool. This parallel transformation allows for more precise extraction of rich features, while fusion improves data classification by efficiently capturing diverse signal characteristics. The sequential nature of LSTM is utilized for the classification of various diseases. To assess the impact of hybrid features from the CAE latent space features from both CWT and the mel spectrogram, we conducted an ablation study using an eight-class classification framework. The results of training the LSTM network with various feature sets are shown in Table 10. When solely CAE latent space features of CWT were used, the LSTM model achieved an average accuracy of 78.50%, an F1-score of 82.14%, a precision of 85.34%, and a sensitivity of 80.42%. In contrast, training with only latent space features from the mel spectrogram resulted in an average accuracy of 90.83%, an F1-score of 85.7%, a precision of 88.31%, and a sensitivity of 84.59%. However, the model’s performance significantly improved when combining both the CAE latent space features, with the accuracy rising to 94.69%, F1-score to 90.69%, precision to 91.89%, and sensitivity to 89.78%. This shows that the fusion of both CAE latent space features significantly improves the LSTM network’s capacity to classify and detect various respiratory disorders in multi-class problems. Table 11 illustrates the overall performance of our proposed model in multiple-class tasks using a publicly available respiratory disease dataset.

The overall accuracy, sensitivity, specificity, and F1-score for the eight-class problems were 94.16%, 89.56%, 99.10%, and 89.5%, respectively. Similarly, for the four-class problems, the overall results were 79.61%, 78.55%, 92.49%, and 78.67%, respectively, and for the three-class problems, the overall results were 89.45%, 88.16%, 94.54%, and 88.41%, respectively. Meanwhile, for the binary-class problems, the overall results for normal vs. abnormal were 85.61%, 83.44%, 83.44%, and 84.21% for accuracy, sensitivity, specificity, and F1-score, respectively, and for crackles and wheezes, they were 84.21%, 93.57%, 93.53%, and 93.15%, respectively. To further validate the robustness of our framework, we also conducted experiments using another public dataset, the SJTU Paediatric dataset [53], for various respiratory diseases, including healthy samples and seven distinct lung diseases: coarse crackle (C), fine crackle (F), rhonchi (R), stridor (S), wheeze (W), and both wheeze and crackle (B). The results, presented in Table 11, demonstrate that our findings are not only applicable to a single dataset but also generalize well across different datasets. This additional validation underscores the generalizability of our model, reinforcing its effectiveness on diverse datasets. The variations in the error rates are associated with the imbalanced nature of the dataset, where some classes are over-represented, influencing the model’s learning bias. Furthermore, the inherent acoustic similarities across various respiratory disorders make it more complex for the model to correctly identify the lung sound. For example, high-pitched sounds like crackles and wheezes provide a special problem since their slight acoustic variances are hidden behind similar spectral sequences.

Several experiments were performed to optimize the proposed model. Specifically, performance was evaluated while varying the learning rate and the number of epochs. Figure 9 shows the classification accuracies across different learning rates ranging from 0.00001 to 0.01 over 200 epochs. The results indicate that for the binary-class problems, the accuracy remained high as the learning rate increased from 0.00001 to 0.001. For the three-class problems, a slight decline in accuracy was observed as the learning rate increased. For the four-class problems, increasing the learning rate noticeably reduced the model’s accuracy after the initial increase, and for the eight-class problems, increasing the learning rate to 0.001 gradually increased the accuracy. Figure 9 indicates that a learning rate of 0.001 over 200 epochs achieved the highest scores across all classification problems. The hybrid approach, combining DL with digital signal processing techniques such as parallel CAEs and dual scalograms, achieved promising results, even on imbalanced datasets.

## 4. Conclusions and Future Work

Our study introduced an advanced, intelligent, lung sound recognition framework for detecting respiratory diseases. We applied dual transformation using mel scalograms and continuous wavelet transform to generate detailed time-frequency scalograms. Parallel convolutional autoencoders were trained to extract essential features from CWT and mel samples. This framework integrates parallel convolutional autoencoders and an LSTM network, reducing the possibility of misclassifying significant features while extracting rich features. The features extracted from both latent spaces are concatenated into a hybrid feature pool and processed through the LSTM model, addressing multiple-class problems. We evaluated our method on the ICBHI 2017 dataset, and the experimental results showed that our proposed model achieved promising results across multiple classification problems. For eight-class problems involving healthy samples and seven distinct lung diseases (asthma, bronchiectasis, bronchiolitis, COPD, LRTI, pneumonia, and URTI), the proposed model achieved an average accuracy of 94.16%, an average sensitivity of 89.56%, an average specificity of 99.10%, and an average F1-score of 89.56%. For the four-class problems, including crackles, wheezes, no label, and both crackles and wheezes, the model achieved an average accuracy of 79.61%, an average sensitivity of 78.55%, an average specificity of 92.49%, and an average F1-score of 78.67%. The results for the three-class problems were an average accuracy of 89.45%, an average sensitivity of 88.16%, an average specificity of 94.54%, and an average F1-score of 88.41%. Finally, for the normal vs. abnormal binary-class problems, the model achieved an average accuracy of 85.61%, an average sensitivity of 83.44%, an average specificity of 83.44%, and an average F1-score of 84.21%, outperforming all other research. In future work, we will deploy the proposed framework in a clinical setting. Additionally, we plan to enhance the robustness of the framework by increasing the number of sound samples through the integration of multiple datasets.

## Figures and Tables

**Figure 1 bioengineering-11-00586-f001:**
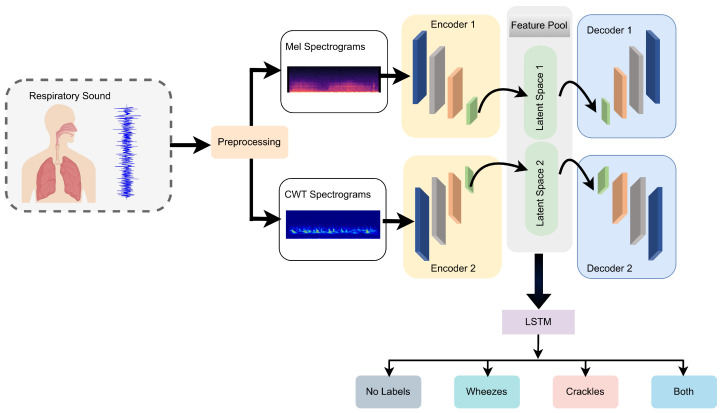
Framework of the proposed method.

**Figure 2 bioengineering-11-00586-f002:**
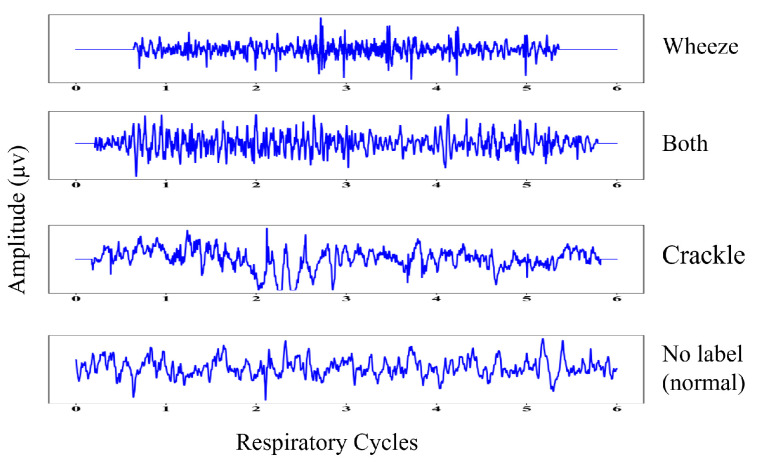
Sample signals of lung sounds.

**Figure 3 bioengineering-11-00586-f003:**
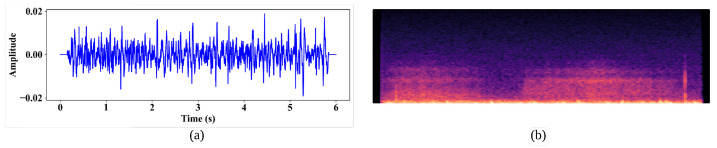
(**a**) The respiratory audio signal, and (**b**) the mel spectrogram respiratory audio signal.

**Figure 4 bioengineering-11-00586-f004:**
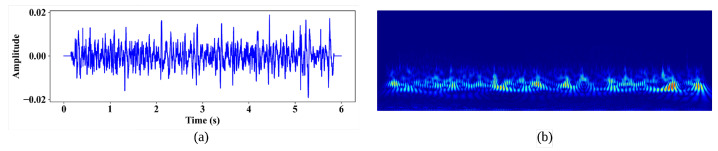
(**a**) The respiratory audio signal, and (**b**) the CWT spectrogram of the respiratory audio signal.

**Figure 5 bioengineering-11-00586-f005:**
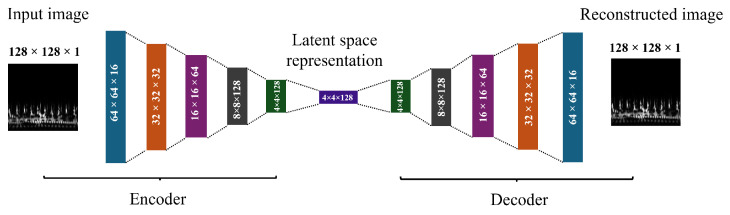
The convolutional autoencoder model architecture.

**Figure 6 bioengineering-11-00586-f006:**
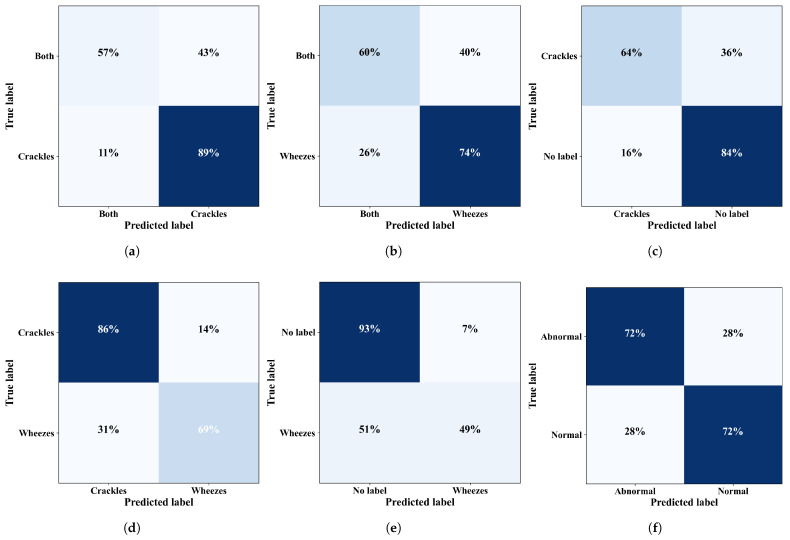
Confusion matrices for binary-class problems: (**a**) B-C, (**b**) B-W, (**c**) C-N, (**d**) C-W, (**e**) N-W, and (**f**) N-Ab.

**Figure 7 bioengineering-11-00586-f007:**
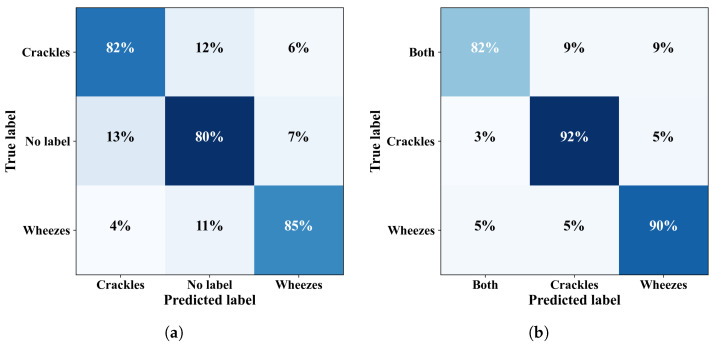
Confusion matrices for three-class problems: (**a**) C-N-W, and (**b**) B-C-W.

**Figure 8 bioengineering-11-00586-f008:**
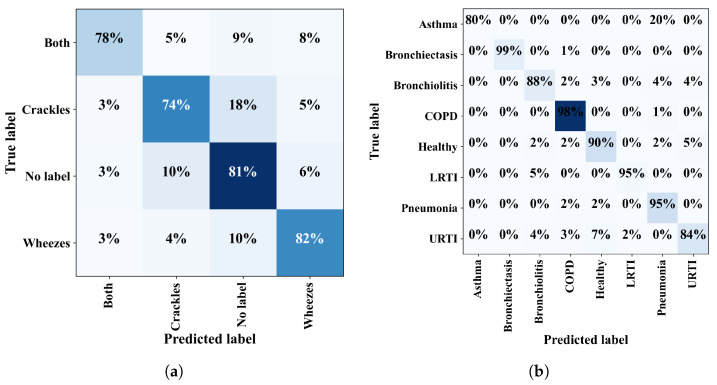
Confusion matrices for (**a**) four-class problems, and (**b**) eight-class problems.

**Figure 9 bioengineering-11-00586-f009:**
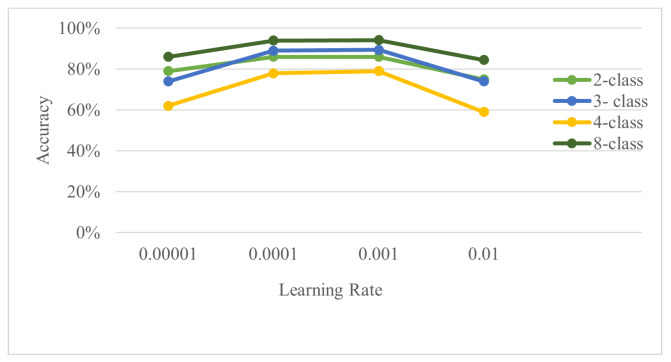
Impact of the learning rate on accuracy.

**Table 1 bioengineering-11-00586-t001:** Description of the dataset.

Cycle	Start Time (s)	End Time (s)	Crackles	Wheezes
01	1.018	3.411	1	0
02	3.411	5.827	1	0
03	5.827	8.339	1	1
04	8.339	10.923	1	0
05	10.923	13.292	0	1
06	13.292	16.018	1	0
07	16.018	18.482	1	0
08	18.482	19.542	1	0

**Table 2 bioengineering-11-00586-t002:** Respiratory cycles of lung sounds from ICBHI-2017.

Sound	Original	Augmented
Crackle	1864	4150
Wheeze	886	3544
No label	3642	5666
Both sounds	506	2024
Total	6898	15,384

**Table 3 bioengineering-11-00586-t003:** Respiratory cycles of lung sounds from ICBHI-2017 for distinct lung diseases.

Disease	Original	Augmented
Asthma	06	24
Bronchiectasis	104	416
Bronchiolitis	160	640
COPD	3642	5746
Healthy	322	1288
LRTI	32	128
Pneumonia	285	1140
URTI	243	972
Total	6898	10,354

**Table 4 bioengineering-11-00586-t004:** The convolutional autoencoder layers.

Encoder		Decoder	
**Layer**	**Output**	**Layer**	**Output**
Conv2D	128 × 128 × 1	Dense	2048
Conv2D	64 × 64 × 16	Reshape	4 × 4 × 128
Conv2D	32 × 32 × 32	Conv2DTranspose	8 × 8 × 128
Conv2D	16 × 16 × 64	Conv2DTranspose	16 × 16 × 64
Conv2D	8 × 8 × 128	Conv2DTranspose	32 × 32 × 32
Conv2D	4 × 4 × 128	Conv2DTranspose	64 × 64 × 16
Flatten	2048	Conv2DTranspose	128 × 128 × 1
Trainable parameters: 147,842			
Non-trainable parameters: 0			
Total parameters: 147,842			

**Table 5 bioengineering-11-00586-t005:** The LSTM architecture.

Layer Type	Output Shape	Parameters
lstm (LSTM)	(None, 64)	147,712
dense 4 (dense)	(None, 2)	130
Total parameters: 147,842		
Trainable parameters: 147,842		
Non-trainable parameters: 0		

**Table 6 bioengineering-11-00586-t006:** Performance metrics for binary-class problems.

Class	Accuracy	F1-Score	Precision	Sensitivity
Non-augmented dataset				
C-W	≈81%	≈78%	≈78%	≈78%
B-C	≈81%	≈68%	≈69%	≈67%
B-W	≈69%	≈67%	≈67%	≈68%
C-N	≈78%	≈75%	≈75%	≈74%
W-N	≈85%	≈73%	≈76%	≈72%
N-Ab	≈67%	≈67%	≈67%	≈67%
Augmented dataset				
C-W	≈94%	≈94%	≈94%	≈94%
B-C	≈94%	≈94%	≈94%	≈94%
B-W	≈94%	≈94%	≈93%	≈94%
C-N	≈86%	≈88%	≈86%	≈85%
W-N	≈90%	≈89%	≈89%	≈99%
N-Ab	≈86%	≈84%	≈85%	≈84%

**Table 7 bioengineering-11-00586-t007:** Performance metrics for three-class problems.

Class	Accuracy	F1-Score	Precision	Sensitivity
Non-Augmented dataset				
B-C-W	≈71%	≈63%	≈64%	≈62%
C-N-W	≈67%	≈60%	≈61%	≈53%
Augmented dataset				
B-C-W	≈90%	≈89%	≈89%	≈88%
C-N-W	≈82%	≈82%	≈82%	≈83%

**Table 8 bioengineering-11-00586-t008:** Performance metrics for four-class problems.

Class	Dataset	Accuracy	F1-Score	Precision	Sensitivity
Four-Class	Non-augmented	≈64%	≈54%	≈55%	≈53%
Four-Class	Augmented	≈80%	≈79%	≈79%	≈79%

**Table 9 bioengineering-11-00586-t009:** Performance metrics for eight-class problems.

Class	Dataset	Accuracy	F1-Score	Precision	Sensitivity
Eight-Class	Non-augmented	≈93%	≈61%	≈61%	≈63%
Eight-Class	Augmented	≈94%	≈90%	≈90%	≈90%

**Table 10 bioengineering-11-00586-t010:** Ablation experiment.

Features	Accuracy %	F1-Score %	Precision %	Sensitivity %
CWT latent space features	87.78	82.14	85.34	80.42
Mel latent space features	90.83	85.72	88.31	84.59
Combined features	94.69	90.68	91.89	89.78

**Table 11 bioengineering-11-00586-t011:** Comparison between the proposed model and already established works.

			Performance %			
Study	Class	Method	Accuracy	Sensitivity	Specificity	F1-Score
Demir et al. [32]	4	CNN, LDA	≈71	≈61	≈86	≈65
Lie et al. [44]	4	ARCS-NET	-	≈41	≈67	≈57
	2		≈80	≈81	≈80	-
Petmezas et al. [33]	4	CNN-LSTM	≈76	≈53	≈85	≈69
Demir et al. [45]	4	VGG-16, SVM	≈66	≈53	≈83	≈55
Ma et al. [46]	4	Bi-ResNet	≈53	≈31	≈69	≈50
	2		-	≈48	≈69	≈49
Chambres et al. [47]	4	HMM, NLSp	≈50	≈21	≈78	≈50
	2		-	≈33	≈78	≈56
Rocha et al. [48]	4	LDA	≈61	≈52	≈66	≈59
Acharya and Basu [15]	4	CNN, RNN	-	≈49	≈84	≈67
Mang et al. [49]	4	Cochleagram, CNN	≈63	≈53	≈69	≈61
Wanasinghe et al. [50]	6	Mel, MFCC, CNN	≈93	≈92	≈98	≈93
Choi et al. [51]	10	Mel, CNN	≈90	-	-	-
Li et al. [52]	4	TQWT, STFT	-	≈37	≈72	≈54
	2		-	≈52	≈72	≈62
**Proposed Model**						
	A-B1-C-U-L-B-P-H		≈94	≈90	≈99	≈90
	C-W-N-B		≈80	≈79	≈93	≈79
	B-C-W		≈90	≈88	≈95	≈89
ICBHI dataset	C-N-W	CAE, LSTM	≈82	≈83	≈91	≈82
	N-AB		≈86	≈84	≈84	≈85
	C-W		≈94	≈95	≈94	≈94
	B-C		≈95	≈94	≈94	≈94
SJTU dataset	C-F-N-R-S-W-B	CAE, LSTM	≈82	≈39	≈92	≈41
	N-AB		≈84	≈75	≈75	≈76

## Data Availability

The datasets that support the findings of this study are openly available and are as follows. Rocha, B.; Filos, D.; Mendes, L.; Vogiatzis, I.; Perantoni, E.; Kaimakamis, E.; Natsiavas, P.; Oliveira, A.; Jácome, C.; Marques, A.; 507 et al. “A respiratory sound database for the development of automated classification” (Accessed on 31 January 2024).

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
