# Peer review of "Auscultation-Based Pulmonary Disease Detection through Parallel Transformation and Deep Learning"

_bioengineering, 2024, doi:10.3390/bioengineering11060586_

Round 1
Reviewer 1 Report
Comments and Suggestions for Authors
This is a truly interesting paper.
The underlying problem is clearly of interest.
The methodology used are well presented.
Suggestions:
Can you better describe the diffentent usefulness of the two time-frequency approaches here used? Are the info gathered somehow redundant?
Why do not consider a CNN approach with a volume input by using both images in the t-f domain as a direct input and not as two separate channels?
The errors made by the models are different in different couples of classes. Are there any reasomable explanation for this that can be done in the Discussion section?
The detection of pulmonary sound can take advantage of a different transform approach based on Empirical Mode decomposition, and possibly using ECG single channel signals; can you argue on this? Consider for this the following papers:
W. Yi and K. Park, "Derivation of respiration from ECG measured without subjects awareness using wavelet transform", Proc. 2nd Joint EMBS/BMES Conf., pp. 130-131, 2002
C. OBrien and C. Heneghan, "A comparison of algorithms for estimation of a respiratory signal from the surface electrocardiogram", Computers in Biology and Medicine, vol. 37, pp. 305-314, 2007
M. Campolo, et al., "ECG-derived respiratory signal using Empirical Mode Decomposition," 2011 IEEE International Symposium on Medical Measurements and Applications, Bari, Italy, 2011, pp. 399-403
Comments on the Quality of English Language
Just some minor corrections are needed
Author Response
The response to the reviewer's questions are presented in the attached file. We request the reviewer to see the attached file.

Reviewer 2 Report
Comments and Suggestions for Authors
Dear author,
I would like to express my gratitude for the research results you have submitted. Your work has provided valuable insights into the diagnosis of lung diseases. Here are some suggestions to help you further improve the quality of your manuscript.
(1) The fonts of the graphs in the article are inconsistent; please ensure that the font styles of all graphs in the article are consistent to enhance the uniformity and professionalism of the visual presentation.
(2) The article lists a large number of major contribution points, but seems to lack in-depth argumentation and explanation. It is recommended to streamline the contribution points and provide a more detailed discussion of each point to ensure that the reader can clearly understand the value and importance of each contribution.
(3) There is an overlap in content between the second and third points of the main contribution. In order to improve the clarity and logic of the article, it is recommended that these two points be merged and their distinctive contributions be clearly articulated.
(4) The three time-domain audio data enhancement methods mentioned in the article need more detailed descriptions and explanations. It is recommended that more technical details be provided, including how the methods work, the expected results, and a comparison with existing techniques.
(5) The article currently uses only a single dataset for experimental validation. To enhance the generalizability and robustness of the findings, it is recommended to introduce more diverse datasets for testing and to explore the performance of the model on different datasets.
(6) Although the article provides rich results of model experiments, it lacks comparative experiments with other methods or models. It is recommended that a comparative analysis be conducted to verify the superiority and validity of the proposed method and to ensure the reliability of the results.
Comments on the Quality of English LanguageDear author,
The language used in the manuscript is generally understandable, yet it could benefit from further refinement to meet the publication's standards.
Author Response

(The authors gave the same response as above.)
